# Growth and Migration Blocking Effect of Nanaomycin K, a Compound Produced by *Streptomyces* sp., on Prostate Cancer Cell Lines In Vitro and In Vivo

**DOI:** 10.3390/cancers15102684

**Published:** 2023-05-10

**Authors:** Yuto Hirata, Katsumi Shigemura, Michika Moriwaki, Masato Iwatsuki, Yuki Kan, Tooru Ooya, Koki Maeda, Youngmin Yang, Takuji Nakashima, Hirotaka Matsuo, Jun Nakanishi, Masato Fujisawa

**Affiliations:** 1Department of Public Health, Kobe University Graduate School of Health Sciences, 7-10-2 Tomogaoka, Suma-Ku, Kobe 654-0142, Japan; 221k319k@cloud.kobe-u.jp (Y.H.);; 2Department of Urology, Kobe University Graduate School of Medicine, 7-5-2 Kusunoki-Cho, Chuo-Ku, Kobe 650-0017, Japan; 3Department of Medical Device Engineering, Graduate School of Medicine, 7-5-2 Kusunoki-Cho, Chuo-Ku, Kobe 650-0017, Japan; 4Center for Advanced Medical Engineering Research & Development (CAMED), Kobe University, 1-5-1 Minatojima-minamimachi, Chuoku, Kobe 657-0047, Japan; 5Ōmura Satoshi Memorial Institute, Kitasato University, 5-9-1 Shirokane, Minato-Ku, Tokyo 108-8641, Japan; 6Graduate School of Infection Control Sciences, Kitasato University, 5-9-1 Shirokane, Minato-Ku, Tokyo 108-8641, Japan; 7Graduate School of Engineering Kobe University, 1-1 Rokkodai-Cho, Nada-Ku, Kobe 657-8501, Japan; 8Research Organization for Nano and Life Innovation, Waseda University, 513 Waseda Tsurumakicho, Shinjuku-Ku, Tokyo 162-0041, Japan; 9Research Center for Medicinal Plant Resources, National Institutes of Biomedical Innovation, Health and Nutrition, 1-2 Hachimandai, Tsukuba 305-8043, Japan; 10Research Center for Macromolecules and Biomaterials, National Institute for Materials Science (NIMS), 1-1 Namiki, Tsukuba 305-0044, Japan

**Keywords:** nanaomycin K, *streptomyces*, prostate cancer, castration-resistant prostate cancer, tumor growth, migration, epithelial mesenchymal transition, MAPK signaling pathway

## Abstract

**Simple Summary:**

Recent molecularly targeted drugs used to treat castration-resistant prostate cancer (CRPC) soon lose effectiveness as CRPC develops resistance to these therapeutics. New molecularly targeted drugs for effective treatment of CRPC are needed. In this study, we investigated the anticancer activity of nanaomycin K, a novel compound extracted from *Streptomyces* sp., in CRPC and non-CRPC cell lines. Nanaomycin K significantly inhibited the growth of CRPC and non-CRPC cells by inducing apoptosis through the Caspase-3 pathway. Nanaomycin K also significantly inhibited migration of CRPC, decreasing its invasive potential. The inhibition of migration by nanaomycin K was shown to be mediated by inhibition of Ras, Slug, and MAPK phosphorylation. In vivo, nanaomycin K also significantly and safely inhibited the growth of tumors derived from CRPC. Nanaomycin K’s anti-tumor effects on CRPC are achieved in part by inhibiting growth and migration.

**Abstract:**

Since castration-resistant prostate cancer (CRPC) acquires resistance to molecularly targeted drugs, discovering a class of drugs with different mechanisms of action is needed for more efficient treatment. In this study, we investigated the anti-tumor effects of nanaomycin K, derived from “*Streptomyces rosa* subsp. *notoensis*” OS-3966. The cell lines used were LNCaP (non-CRPC), PC-3 (CRPC), and TRAMP-C2 (CRPC). Experiments included cell proliferation analysis, wound healing analysis, and Western blotting. In addition, nanaomycin K was administered intratumorally to TRAMP-C2 carcinoma-bearing mice to assess effects on tumor growth. Furthermore, immuno-histochemistry staining was performed on excised tissues. Nanaomycin K suppressed cell proliferation in all cell lines (*p* < 0.001) and suppressed wound healing in TRAMP-C2 (*p* = 0.008). Nanaomycin K suppressed or showed a tendency to suppress the expression of N-cadherin, Vimentin, Slug, and Ras in all cell lines, and suppressed the phosphorylation of p38, SAPK/JNK, and Erk1/2 in LNCaP and TRAMP-C2. In vivo, nanaomycin K safely inhibited tumor growth (*p* = 0.001). In addition, suppression of phospho-Erk1/2 and increased expression of E-cadherin and cleaved-Caspase3 were observed in excised tumors. Nanaomycin K inhibits tumor growth and suppresses migration by inhibiting epithelial-mesenchymal transition in prostate cancer. Its mechanism of action is related to the inhibition of phosphorylation of the MAPK signaling pathway.

## 1. Introduction

Prostate cancer (PCa) is the second most frequent cancer and the fifth leading cause of cancer death in men [1]. Metastatic PCa has a high degree of malignancy, and hormone therapy aimed at suppressing androgen secretion and activity is the first-line treatment. However, PCa can acquire castration resistance and become castration-resistant prostate cancer (CRPC) within a few years [2]. Although docetaxel and other drugs are currently approved for CRPC, the side effects are strong, continuous administration is difficult, and further resistance may be acquired [3]. There is a need for new therapeutic agents with different mechanisms of action and fewer side effects.

One of the mechanisms related to PCa’s acquisition of infiltration ability is epithelial-mesenchymal transition (EMT). EMT is the process by which epithelial cells acquire the traits of mesenchymal cells by reducing cell–cell adhesion, degrading the basement membrane, losing cell polarity, and progressing to migration and infiltration [3]. EMT also contributes to cancer metastasis as cancer cells infiltrate blood vessels and lymph vessels. Thus, inhibiting EMT can inhibit PCa metastasis.

EMT is defined by decreased expression of epithelial markers such as E-cadherin and increased expression of mesenchymal markers such as N-cadherin and vimentin [4]. Various signaling pathways such as transforming growth factor beta (TGF-β), Wnt, Notch, and Hedgehog are known to be involved in EMT [5]. These signaling pathways ultimately induce EMT-inducing transcription factors such as Slug, Snail, ZEB1/2, and TWIST1/2 [6].

EMT also promotes cancer stem cell transformation and is strongly associated with resistance to therapeutic drugs [5]. It has also been implicated in ADT resistance in prostate cancer [7]. Suppressing EMT is therefore important in the treatment of prostate cancer, as EMT is involved in both metastasis and the induction of drug resistance.

Currently, PARP inhibitors such as Olaparib and Rucaparib are used for the treatment of CRPC. These molecularly targeted drugs induce cancer cell death by inhibiting Poly (ADP-ribose) polymerase-1 (PARP-1), which promotes the repair of DNA single-strand breaks [8]. PARP inhibitors are expected to have anti-tumor effects in PCa where PARP-1 is overexpressed [9]. However, BRCA1/2 gene mutations are required for PARP inhibitors to be effective, and the incidence of BRCA1/2 gene mutations in metastatic CRPC patients is only about 6.2% [10]. Therefore, molecularly targeted drugs with other mechanisms are needed to treat CRPC without BRCA1/2 gene mutations.

Nanaomycin is a natural compound found in the culture medium of “*Streptomyces rosa* subsp. *notoensis*” OS-3966. Currently, 12 analogues of nanaomycins A–K have been discovered (Figure 1). Nanaomycins A–E have been found to be anti-mycoplasma substances, and nanaomycin A has been used as a treatment for bovine dermatophytosis [11,12,13]. Nanaomycins F–K were found to inhibit the proliferation of EMT-induced cells [14]. In particular, nanaomycin K is produced in higher quantities than nanaomycin H [14], and nanaomycin K has shown anti-tumor effects on bladder cancer cell lines [15].

In this study, we examined the EMT-inhibitory effect of nanaomycin K on PCa in vitro and in vivo using the human androgen-dependent LNCaP and androgen-independent PC-3 PCa cell lines as well as the mouse androgen-independent PCa cell line TRAMP-C2.

## 2. Materials and Methods

### 2.1. Cells and Reagents

Two human PCa cell lines, androgen-dependent LNCaP and androgen-independent PC-3, were grown in RPMI-1640 medium containing 10% fetal bovine serum (FBS) (Sigma-Aldrich, St. Louis, MO, USA) and 1% penicillin/streptomycin (P/S) (FUJIFILM Wako Pure Chemicals, Osaka, Japan) at 37 °C and 5% CO_2_. Murine androgen-independent PCa cell line TRAMP-C2 was maintained in D-MEM medium with 10% FBS and 1% P/S at 37 °C and 5% CO_2_. Nanaomycin K was obtained from the culture broth of “*Streptomyces rosa* subsp. *notoensis*” OS-3966 using the previously described purification methods [14]. Nanaomycin K was dissolved in dimethyl sulfoxide (DMSO) and then diluted.

### 2.2. Cell Proliferation Assays

We conducted cell proliferation assays with LNCaP, PC-3, and TRAMP-C2 in the presence of nanaomycin K to investigate its anti-tumor activity in vitro. One thousand LNCaP, PC-3, and TRAMP-C2 cells were seeded and incubated for 24 h before being divided into two groups. One group was placed in media containing 5 ng/mL of TGF-β (FUJIFILM Wako Pure Chemicals), while the other group was placed in media without it [16,17]. After switching media, 5 µg/mL nanaomycin K or DMSO was added to the cultures. Cell proliferation was assessed at 0, 24, 48, and 72 h of incubation using 3-(4,5-dimethylthiazol-2-yl)-5-(3-carboxymethoxyphenyl)-2-(4-sulfophenyl)-2H-tetrazolium (MTS) (Promega Corporation, Madison, WI, USA). All experiments were carried out in triplicate.

### 2.3. Wound Healing Assays

To assess the anti-tumor activity of nanaomycin K in vitro, wound healing assays were conducted using LNCaP, PC-3, and TRAMP-C2 cells. One hundred thousand cells were seeded and incubated overnight, after which they were divided into two groups and treated with media containing or lacking 5 ng/mL TGF-β for 24 h. Following incubation, nanaomycin K (10 µg/mL) or 0.10% DMSO was added to the culture. The cell monolayers were scratched using 200 µL micropipette tips in each well after they had been incubated for 48 h. Then, cells were washed and fresh medium was added. Microscopic images were obtained at three intervals after the cells were scratched: at the start, 6 h, and 12 h [15]. All experiments were carried out in triplicate.

### 2.4. Western Blotting

One hundred thousand cells were seeded and incubated overnight, after which they were divided into two groups and treated with media containing or lacking 5 ng/mL TGF-β. A 24 h incubation period was followed by the addition of nanaomycin K (25 µg/mL) or 0.25% DMSO to the cultures. After cells were incubated for an additional 48 h in the presence of nanaomycin K or DMSO, they were washed and lysed in 8 M urea buffer. The sample buffer (Nacalai Tesque, Kyoto, Japan) was heated at 95 °C for 5 min and then combined with each sample. The samples were separated using SDS-PAGE and then transferred onto PVDF membranes. After blocking with Blocking One (Nacalai Tesque) or Blocking One-P (Nacalai Tesque) and washing, the membranes were left to incubate overnight at 4 °C with antibodies against E-cadherin (Biolegend, Hsinchu City, Taiwan), N-cadherin (Proteintech, Rosemont, IL, USA), vimentin (Proteintech), phospho-p38 MAPK (Thr180/Tyr182) (Cell Signaling Technology: CST, Danvers, MA, USA), phospho-SAPK/JNK (Thr183/Tyr185) (CST), phospho-p44/42 MAPK (Erk1/2) (Thr202/Tyr204) (CST), Snail (CST), Slug (CST), Ras (CST), or β-actin (Santa Cruz Biotechnology, Dallas, TX, USA). After the membranes had been washed again, they were incubated with HRP-conjugated secondary antibodies (Anti-IgG (H + L chain) (Mouse) pAb-HRP or Anti-IgG (H + L chain) (Rabbit) pAb-HRP (MBL, Nagoya, Japan)) for one hour at room temperature. Protein–antibody binding was then detected using enhanced chemiluminescence.

### 2.5. Animal Experiments

To investigate the in vivo anti-tumor effects of nanaomycin K, animal experiments were conducted using a mouse prostate cancer model. Male C57BL/6J mice, aged 6–8 weeks, were obtained from CLEA Japan, Inc (Tokyo, Japan). One million cells were inoculated subcutaneously at day 0 (*n* = 5, respectively) with VitroGel 3D (TheWell Bioscience, NJ, USA). After the tumor’s long axis exceeded 10 mm, mice were randomly assigned to the treatment group (0.5 mg/mouse and 1.0 mg/mouse of nanaomycin K) or the control group (DMSO). The solvent used for administration to the animals was PBS, with a dosage of 80 μL. The proportion of DMSO administered to the control group was 12%. Nanaomycin K was intratumorally injected with Spongel (LTL Pharma, Tokyo, Japan). The size of the tumor was calculated using the following formula: (longest diameter) × (shortest diameter)^2^ × 0.5. The mice were terminated and tumors were harvested after five days of treatment [16]. 

### 2.6. Immunohistochemical Staining

Tissue sections were prepared by embedding fixed tumor tissue in paraffin, then dewaxing and rehydrating them. Antigen retrieval was performed by heating the sections in citrate buffer (pH 6.0 or 9.0) at 98 °C for 20 min. Immunohistochemical staining (IHC) was performed on the tissue sections using an automatic tissue processor (Bond-Max; Leica Microsystems, Wetzlar, Germany) according to the standard protocol. Briefly, the sections were incubated with primary antibodies anti-E-cadherin (Proteintech), anti-phospho-Erk1/2 (Biolegend), and anti-cleaved-Caspase 3 (CST). The sections were then treated with HRP-conjugated secondary antibody (BOND Polymer Refine Detection (Leica)) according to the standard protocols of the instrument after washing. The tissue sections were stained with diaminobenzidine and counterstained with hematoxylin. The resulting tissue slides were examined using a BZ-X710 microscope (Keyence, Osaka, Japan).

### 2.7. Immunohistochemical Analysis

Based on the proportion of positive cells, IHC scoring calculated the staining intensity as follows: 0 (negative), 1+ (weak), 2+ (medium), or 3+ (strong). The percentage of stained cells (frequency score) was divided into three categories: 1, 0–10%; 2, 11–50%; and 3, more than 50% stained cells. The frequency and intensity scores were multiplied to arrive at the IHC score. This was performed for five fields of view for each group, and the average of these was used as the final IHC score [16].

### 2.8. Ethical Approval

All animal studies were carried out in accordance with institutional ethical standards, the ARRIVE guidelines, and all pertinent rules and regulations. No author has ever conducted any research using human subjects. The Kobe University institutional ethics and animal welfare committees reviewed and gave their approval to every aspect of the experimental design and procedure.

### 2.9. Statistical Analysis

Comparisons between two different groups were made using Student’s *t*-test. Statistical differences between means were regarded as significant when *p* < 0.05 was reached.

## 3. Results

### 3.1. Nanaomycin K Inhibited the Growth of Prostate Cancer Cells

Absorbance at 0 h after the addition of 1.5 µg/mL of nanaomycin K significantly inhibited LNCaP, PC-3, and TRAMP-C2 cancer cell growth compared to control cells after 24 h of culture (*p* < 0.05, or *p* < 0.01) (*n* = 3, Student’s *t*-test) (Figure 2). Nanaomycin K at higher concentrations demonstrated strong cell cytotoxicity in vitro. In LNCaP, 5 µg/mL nanaomycin K significantly inhibited cell proliferation in the presence of TGF-β.

### 3.2. Migration-Inhibitory Effect of Nanaomycin K

Wound healing assays investigated whether cell migration was affected by nanaomycin K. In LNCaP, nanaomycin K inhibited wound closure, but the difference was not significant. On the other hand, nanaomycin K significantly inhibited cell closure in TRAMP-C2 12 h after scratching the culture, in the presence of TGF-β (*p* = 0.008) (*n* = 3, Student’s *t*-test) (Figure 3).

### 3.3. Expression of EMT-Related Protein and MAPK Signaling after Culture with Nanaomycin K

Nanaomycin K decreased the expression of N-cadherin in LNCaP, PC-3, and TRAMP-C2, and decreased the expression of Vimentin in LNCaP and PC-3 cells in the absence of TGF-β as well as in LNCaP cells in the presence of TGF-β. A decreasing trend was observed under other conditions, but without a significant difference (Figure 4a). Nanaomycin K appeared to decrease the expression of Slug, a family of transcription factors that induce EMT, in all cell lines, but Snail showed no significant changes caused by 25 µg/mL of nanaomycin K (Figure 4b). Regarding the MAPK signaling pathway, the expression of phospho-p38 and phospho-SAPK/JNK was reduced by nanaomycin K in LNCaP, as was the expression of phospho-ERK1/2 in LNCaP, while a decreasing trend was observed in TRAMP-C2 (Figure 4c). In particular, the expression of phospho-p38 in LNCaP and phospho-ERK1/2 in TRAMP-C2 was suppressed by nanaomycin K in the presence of TGF-β. Nanaomycin K appeared to decrease the expression of Ras in LNCaP and PC-3 in the absence of TGF-β, as well as in PC-3 cells in the presence of TGF-β, while a decreasing trend was observed in TRAMP-C2 in the presence of TGF-β, but without a significant difference (Figure 4d). Full pictures of the Western blots and the densitometry scans are presented in Appendix A.

### 3.4. Nanaomycin K Inhibited Tumor Growth In Vivo

Intratumoral injection of nanaomycin K at both 0.5 mg/body and 1.0 mg/body significantly inhibited TRAMP-C2 tumor growth after 2 days of treatment compared to controls, and the effect of 1.0 mg/body was stronger than that of 0.5 mg/body (*p* = 0.002 and *p* = 0.003, respectively). During treatment days 3–5, the relative tumor volume of the control group increased, while the relative tumor volume of the 0.5 mg/body and 1.0 mg/body group changed little. After 5 days of treatment, the 0.5 mg/body and 1.0 mg/body groups showed significantly inhibited tumor formation compared to controls (both *p* = 0.001, respectively) (*n* = 5, Student’s *t*-test) (Figure 5). There was no significant difference between 0.5 mg/body and 1.0 mg/body. No negative side effects were observed in either group after treatment.

### 3.5. Changes of E-Cadherin, Phosho-ERK1/2 and Cleaved-Caspase-3 in Tumor Tissues after Treatment with Nanaomycin K

In TRAMP-C2, nanaomycin K significantly increased the expression of E-cadherin (0.5 mg/body and 1.0 mg/body: *p* < 0.001) (*n* = 5, Student’s *t*-test). In addition, phospho-Erk1/2 expression was decreased by nanaomycin K treatment compared to control mice. Effects were more pronounced at 1.0 mg/body of nanaomycin K (0.5 mg/body: *p* = 0.089, 1.0 mg/body: *p* = 0.031) (*n* = 5, Student’s *t*-test). The expression of cleaved-Caspase3 was concentration-dependently increased by nanaomycin K (0.5 mg/body: *p* = 0.009, 1.0 mg/body: *p* < 0.001) (*n* = 5, Student’s *t*-test) (Figure 6).

## 4. Discussion

Nanaomycin K is a natural compound found in the cultured broth of “*Streptomyces rosa* subsp. *notoensis*” OS-3966, as a new analog of nanaomycin having an ergothioneine group in its partial structure [14]. Nanaomycin K has been reported to have strong anti-tumor and EMT-inhibitory effects on bladder cancer cell lines [15]. In this study, we evaluated the ability of nanaomycin K to inhibit tumor growth and suppress the process of EMT in vitro and in vivo in CRPC cell lines.

In vitro, nanaomycin K inhibited the growth, migration, and metastasis of prostate cancer cell lines. Nanaomycin K inhibited cell proliferation in the presence or absence of TGF-β. TGF-β is secreted by tumor cells and cells in the tumor microenvironment. In cancers with advanced malignant transformation, increased expression of TGF-β strongly induces cancer cell proliferation [18], and TGF-β is known to be associated with proliferation in prostate cancer [19], suggesting that the growth of prostate cancer cell lines was inhibited by nanaomycin K. Other studies have reported that the relative fold change after 72 h of treatment with Olaparib in PC-3 was 5.1 [20]. The relative fold change after 72 h of treatment with nanaomycin K in PC-3 is expected to show a stronger growth inhibitory effect since it is smaller.

Cancer cell migration is a critical factor in the spread of cancer. Wound healing assays suggested that nanaomycin K inhibits migration of CRPC cell lines stimulated by TGF-β. TGF-β promotes EMT through the TGF-β signaling cascade and enhances the migration ability of cancer cells [21,22,23]. Nanaomycin K appeared to inhibit the EMT migration induced by TGF-β.

With regard to the mechanism of EMT suppression, Western blotting showed that nanaomycin K decreases the expression of the EMT markers N-cadherin and vimentin at the protein level. Prostate cancer progression increases the mesenchymal markers N-cadherin and vimentin; N-cadherin promotes EMT via activation of the ErbB signaling pathway and enhances the migratory and invasive potential of cancer cells [24]. Vimentin is also used as a marker for EMT since it is barely expressed in epithelial cells, but shows increased expression when cells acquire a mesenchymal phenotype, and it promotes EMT by regulating the E-cadherin/β-catenin complex and other functions [25], thus suggesting that nanaomycin K inhibits EMT at the protein level.

Additionally, Western blotting showed that nanaomycin K decreases the expression of Slug transcription factors that promote EMT markers at the protein level. In a model of TGF-β-induced prostatic EMT, Slug is the dominant regulator of EMT initiation [26]. Nanaomycin K may suppress Slug-induced EMT.

The results of cell proliferation and wound healing analyses showed that nanaomycin K was more effective against cell lines in which TGF-β was applied. In this study, we focused on the MAPK signaling pathway activated by TGF-β to determine the mechanism of action, but PI3K is another protein known to be activated by TGF-β [27]. PI3K is involved in cell proliferation, migration, differentiation, and cell death; its abnormal activity in PCa and CRPC has been shown to contribute to cancer malignancy [28,29]. It has been reported that inhibition of both PI3K and MAPK in CRPCs improves the therapeutic efficacy of microtubule-targeting drugs such as docetaxel [30]. Therefore, in the future, we will investigate whether nanaomycin K inhibits PI3K activation and whether the combination of nanaomycin K and PI3K inhibitors will show even better anti-tumor effects.

The MAPK signaling pathway is critical for various cancer processes including EMT, cell proliferation, differentiation, and apoptosis. Park et al. stated that the knockdown of Cathepsin A, which is highly expressed compared to normal prostate tissue and causes the inactivation of p38, a major MAPK signaling pathway protein, has been shown to inhibit proliferation by arresting the cell cycle and migration by suppressing EMT, and anti-tumor effects [31]. Our study also demonstrated that it causes the inactivation of Erk1/2, p38, and JNK, the three main proteins of the MAPK signaling pathway. Thus, nanaomycin K could be a molecularly targeted drug against the MAPK signaling pathway.

In addition, Western blot analysis demonstrated that nanaomycin K reduced the expression of Ras. Ras is an important protein that links PI3K/MAPK signaling [32]. Ras is overexpressed in many cancers and is involved in cancer growth and metastasis through the activation of PI3K/MAPK, making it a potential target for cancer therapy [33]. Ras is known to activate ERK1/2 in the MAPK signaling pathway [34], and it is therefore suggested that the reduction of Ras expression by nanaomycin K may be one of the mechanisms by which it inhibits the activation of the MAPK signaling pathway.

In animal experiments, nanaomycin K had an inhibitory effect on tumor growth compared to controls, without any adverse effects. Our previous study in bladder cancer cell lines showed that nanaomycin K has a significant dose-dependent anti-tumor effect [15]. In this PCa study, no dose-dependence was observed, and the same level of anti-tumor effects was observed. Thus, prostate cancer cell lines may be more sensitive to nanaomycin K than bladder cancer cell lines.

Immunohistochemical analysis suggested that nanaomycin K’s anti-tumor effects in castration-resistant prostate cancer involve suppressing Erk1/2 phosphorylation and inducing apoptosis in vivo. Regarding phosphorylated Erk1/2, in a study of prostate cancer patients, a significant increase was observed in CRPC compared to primary prostate cancer, confirming the association between phosphorylated Erk1/2 and biochemical recurrence [35]. Caspase 3 expression is also decreased in the tissues of patients determined to have higher-grade prostate cancer associated with apoptosis resistance in CRPC [36,37] Thus, nanaomycin K’s therapeutic mechanism of action may involve decreasing Erk1/2 and increasing Caspase 3.

Docetaxel is the current standard of care for CRPC. Although the use of docetaxel improves clinical outcomes and prolongs survival, resistance is acquired in many cases [38], and EMT is involved in the acquisition of resistance to anticancer drugs by promoting cancer stemness and mediating resistance to chemotherapy [39]. EMT is also involved in the acquisition of docetaxel resistance, and suppression of EMT has been reported to significantly increase chemosensitivity to docetaxel [40,41]. Nanaomycin K inhibits EMT via suppression of MAPK pathway activation and has shown no significant adverse effects in animal studies. Therefore, the combination of nanaomycin K with currently used CRPC drugs such as docetaxel may enhance the efficacy of CRPC drugs and contribute to improved clinical outcomes and survival.

It is important to note the limitations of this study. First, we used only three prostate cancer cell lines to evaluate the anti-tumor properties of nanaomycin K. Next, a larger number of samples for wound healing and Western blotting analysis would obtain more definitive results. Next, more detailed studies are needed on the mechanisms that inhibit the EMT and MAPK pathways. Additionally, the number of experimental animals used in in vivo experiments was not large. Furthermore, a positive subject group needs to be added to clarify the effect of nanaomycin K. Moreover, the present study used the TRAMP-C2 cell line as an androgen-independent cell line, as in previous studies [42], but whether this cell line is androgen-dependent or not is controversial and needs to be investigated in the future. Lastly, this study needs more detailed mechanical exploration, a blocking study of the related protein in vivo, pharmacokinetics, an investigation of different routes of administration, and toxicity examinations in nanaomycin K. Such studies will be undertaken in the next paper.

## 5. Conclusions

Our findings suggest that treatment with nanaomycin K resulted in the suppression of MAPK signaling pathway phosphorylation and reduced the growth and migration of prostate cancer cells. Our results indicate that the anti-tumor effects of nanaomycin K may be associated with inhibition of MAPK signaling pathway activation. Additional in vitro and in vivo research on nanaomycin K is necessary.

## Figures and Tables

**Figure 1 cancers-15-02684-f001:**
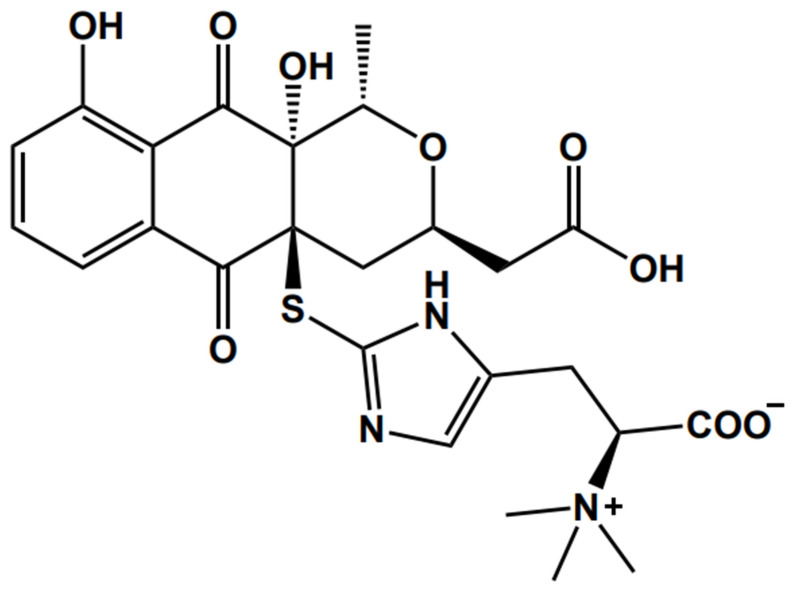
Structure of nanaomycin K.

**Figure 2 cancers-15-02684-f002:**
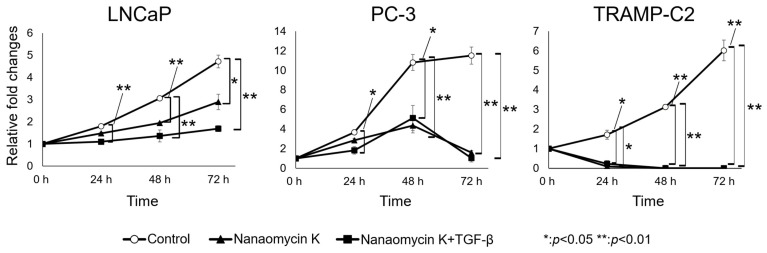
LNCaP, PC-3 and TRAMP-C2 cell proliferation effects of nanaomycin K in vitro. Cell proliferation in LNCaP, PC-3 and TRAMP-C2 cell lines was evaluated in vitro after treating the cells with 5 µg/mL nanaomycin K in the presence or absence of TGF-β for 72 h. Cells treated with DMSO were used as vehicle treated controls (*n* = 3, average ± SE bars, * *p* < 0.05, ** *p* < 0.01). The relative changes in cell proliferation were plotted as a function of time and normalized to the cell proliferation at the beginning of the culture period.

**Figure 3 cancers-15-02684-f003:**
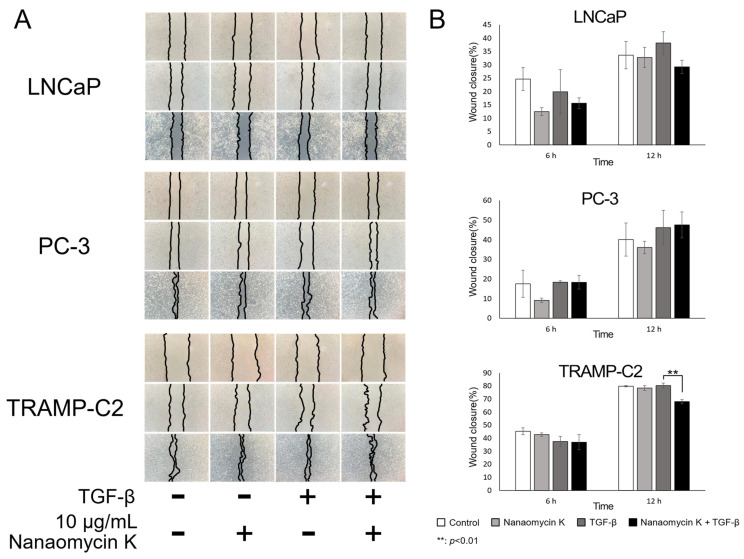
Wound-healing inhibitory effect of nanaomycin K. (**A**) The ability of LNCaP, PC-3, and TRAMP-C2 cells to migrate was studied in the presence of 10 µg/mL nanaomycin K, with or without TGF-β, during a culture period of up to 12 h. (**B**) Wound closures compared to the wound at 0 h (*n* = 3, average ± SE bars, ** *p* < 0.01).

**Figure 4 cancers-15-02684-f004:**
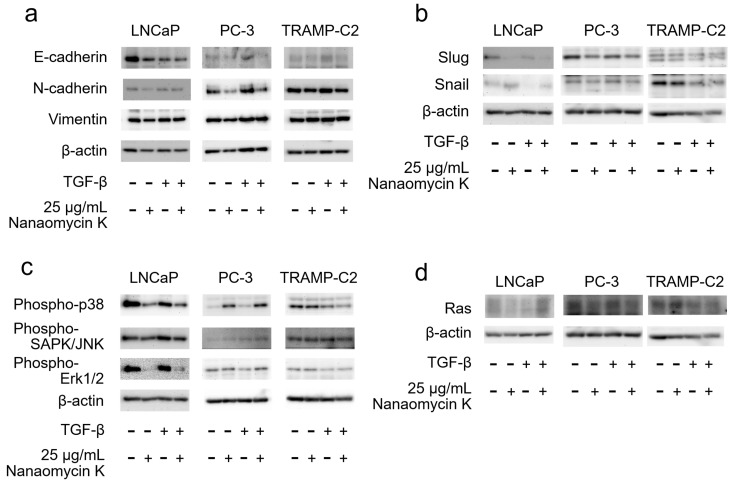
Protein expression of EMT-related markers and MAPK signaling. The expressions of (**a**) EMT markers (E-cadherin, N-cadherin, and Vimentin), (**b**) E-cadherin repressors (Slug, Snail), (**c**) MAPK signaling (phospho-p38, phospho-SAPK/JNK, phospho-ERK1/2), and (**d**) Ras were determined in the presence of 25 µg/mL nanaomycin K and in the presence or absence of TGF-β in vitro for 48 h in LNCaP, PC-3, and TRAMP-C2 cells. β-actin was used as a housekeeping protein.

**Figure 5 cancers-15-02684-f005:**
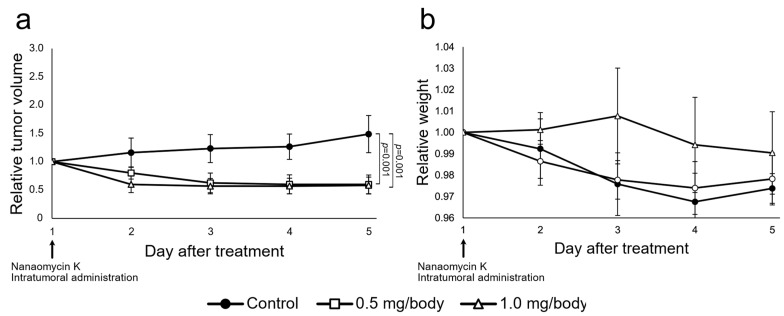
In vivo anti-tumor effects of nanaomycin K. Prostate cancer cell line TRAMP-C2 was subcutaneously inoculated into C57BL/6 mice. On day 1, mice with confirmed tumor growth received intratumoral treatment with either 0.5 mg/body or 1.0 mg/body nanaomycin K or vehicle control. (**a**) Tumor volume was measured for 5 days and standardized to the volume on day 1 to calculate the tumor growth ratio, which is depicted in the graphs (*n* = 5, average ± SE bars). (**b**) Relative mouse weight was measured for 5 days and standardized to the volume on day 1 to calculate the tumor growth ratio, which is depicted in the graphs (*n* = 5, average ± SE bars).

**Figure 6 cancers-15-02684-f006:**
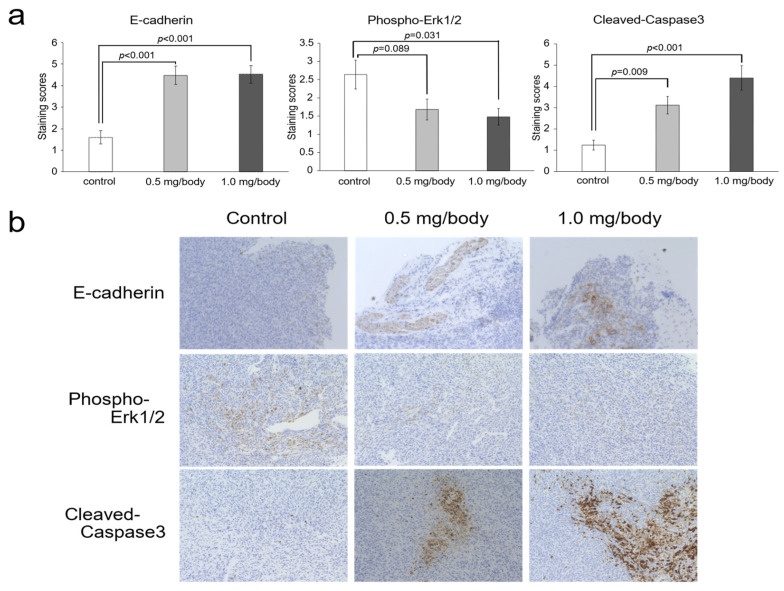
Immunohistochemical analysis of TRAMP-C2 mouse tumors treated with nanaomycin K for *E-cadherin,* phospho-Erk1/2, and cleaved-Caspase 3. (**a**) Immunohistochemical analysis of TRAMP-C2 mouse tumors for E-cadherin, phospho-Erk1/2, and cleaved-Caspase 3 after nanaomycin K treatment. Immunohistochemical staining was used to assess the expression of each marker in tumor tissues, which were then evaluated using a staining score ranging from 0 to 9. (**b**) Representative images for each marker are shown.

## Data Availability

The data is available from the corresponding author upon reasonable request.

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
