# Peer review of "Growth and Migration Blocking Effect of Nanaomycin K, a Compound Produced by Streptomyces sp., on Prostate Cancer Cell Lines In Vitro and In Vivo"

_cancers, 2023, doi:10.3390/cancers15102684_

Round 1

Reviewer 1 Report (Previous Reviewer 3)

Second review report for a manuscript entitled “Anti-tumor Effect of Nanaomycin K, a Compound Produced by Streptomyces sp., on Prostate Cancer Cell Lines In Vitro and In Vivo” submitted by the first author Yuto Hirata and corresponding authors Katsumi Shigemura. The authors ordered many positive data fitting their advocation. However, this study is too immature, and many data are scientifically questionable.

1.         Limited cell line selection: The study focuses on three prostate cancer cell lines (LNCaP, PC-3, and TRAMP-C2), which may not be fully representative of the heterogeneity of prostate cancer. Including additional cell lines or patient-derived models would strengthen the conclusions drawn from the in vitro experiments.

2.         Absence of a positive control: The study does not include a positive control, such as a known inhibitor of the MAPK signaling pathway or a standard-of-care treatment for CRPC, to compare the efficacy of nanomycin K. The inclusion of such controls would help to better contextualize the results and assess the potential therapeutic value of nanomycin K.

3.         Inadequate investigation of molecular mechanisms: The study primarily focuses on the effects of nanomycin K on epithelial-mesenchymal transition (EMT) and the MAPK signaling pathway. However, other molecular mechanisms and signaling pathways that may contribute to the anti-tumor effects of nanomycin K have not been explored. A more in-depth investigation of the drug's molecular targets and the affected pathways would strengthen the study.

4.         Insufficient in vivo evaluation: Although the study includes an in vivo experiment with TRAMP-C2 carcinoma-bearing mice, the evaluation is limited to intratumoral administration of nanomycin K. Investigating different routes of administration, such as oral or intravenous, would provide more information on the drug's bioavailability, pharmacokinetics, and potential clinical applications.

5.         Incomplete assessment of potential toxicities and side effects: The study claims that nanomycin K is safe, but it does not provide a comprehensive evaluation of potential toxicities and side effects associated with the drug. This information is crucial for understanding the safety and feasibility of nanomycin K as a potential treatment for prostate cancer.

6.         Insufficient statistical analysis: The study reports p-values for some experiments, but it lacks a detailed description of the statistical methods used and the number of replicates for each experiment. Providing a comprehensive statistical analysis would enhance the reliability of the study's findings.

7.         Lack of investigation into the drug's pharmacokinetics and pharmacodynamics: The study does not investigate the pharmacokinetics and pharmacodynamics of nanomycin K, which are essential for understanding its absorption, distribution, metabolism, and excretion. These factors are critical for determining the drug's optimal dosing and potential drug-drug interactions in future clinical studies.

Author Response

Second review report for a manuscript entitled “Anti-tumor Effect of Nanaomycin K, a Compound Produced by Streptomyces sp., on Prostate Cancer Cell Lines In Vitro and In Vivo” submitted by the first author Yuto Hirata and corresponding authors Katsumi Shigemura. The authors ordered many positive data fitting their advocation. However, this study is too immature, and many data are scientifically questionable.

  1. Limited cell line selection: The study focuses on three prostate cancer cell lines (LNCaP, PC-3, and TRAMP-C2), which may not be fully representative of the heterogeneity of prostate cancer. Including additional cell lines or patient-derived models would strengthen the conclusions drawn from the in vitro experiments.

[Amendment]

Thank you for comments to improve our manuscript. As suggested, we have added to the study limitation as follows:

“First, we used only three prostate cancer cell lines to evaluate the anti-tumor properties of nanaomycin K.” (line368-369)

  1. Absence of a positive control: The study does not include a positive control, such as a known inhibitor of the MAPK signaling pathway or a standard-of-care treatment for CRPC, to compare the efficacy of nanomycin K. The inclusion of such controls would help to better contextualize the results and assess the potential therapeutic value of nanomycin K.

[Amendment]

Thank you for comments to improve our manuscript. As suggested, we have added to the study limitation as follows:

“Then, a positive subject group needs to be added to clarify the effect of Nanaomycin K.” (line373-374)

  1. Inadequate investigation of molecular mechanisms: The study primarily focuses on the effects of nanomycin K on epithelial-mesenchymal transition (EMT) and the MAPK signaling pathway. However, other molecular mechanisms and signaling pathways that may contribute to the anti-tumor effects of nanomycin K have not been explored. A more in-depth investigation of the drug's molecular targets and the affected pathways would strengthen the study.

[Amendment]

Thank you for comments to improve our manuscript. As suggested, we have added to the study limitation as follows:

“Lastly, this study needs more detailed mechanical exploration, the blocking study of the related protein in vivo, pharmacokinetics, investigation of different routes of administration and toxicity examinations in nanaomycin K. Further such studies will be undertaken in the next paper.” (line378-381)

  1. Insufficient in vivo evaluation: Although the study includes an in vivo experiment with TRAMP-C2 carcinoma-bearing mice, the evaluation is limited to intratumoral administration of nanomycin K. Investigating different routes of administration, such as oral or intravenous, would provide more information on the drug's bioavailability, pharmacokinetics, and potential clinical applications.

[Amendment]

Thank you for comments to improve our manuscript. As suggested, we have added to the study limitation as follows:

“Lastly, this study needs more detailed mechanical exploration, the blocking study of the related protein in vivo, pharmacokinetics, investigation of different routes of administration and toxicity examinations in nanaomycin K. Further such studies will be undertaken in the next paper.” (line378-381)

  1. Incomplete assessment of potential toxicities and side effects: The study claims that nanomycin K is safe, but it does not provide a comprehensive evaluation of potential toxicities and side effects associated with the drug. This information is crucial for understanding the safety and feasibility of nanomycin K as a potential treatment for prostate cancer.

[Amendment]

Thank you for comments to improve our manuscript. As suggested, we have added to the study limitation as follows:

“Lastly, this study needs more detailed mechanical exploration, the blocking study of the related protein in vivo, pharmacokinetics, investigation of different routes of administration and toxicity examinations in nanaomycin K. Further such studies will be undertaken in the next paper.” (line378-381)

  1. Insufficient statistical analysis: The study reports p-values for some experiments, but it lacks a detailed description of the statistical methods used and the number of replicates for each experiment. Providing a comprehensive statistical analysis would enhance the reliability of the study's findings.

[Amendment]

Thank you for comments to improve our manuscript. The number of samples and statistical methods used were added before each p-value. (line206, 221-222, 262, 278, 281, 283)

  1. Lack of investigation into the drug's pharmacokinetics and pharmacodynamics: The study does not investigate the pharmacokinetics and pharmacodynamics of nanomycin K, which are essential for understanding its absorption, distribution, metabolism, and excretion. These factors are critical for determining the drug's optimal dosing and potential drug-drug interactions in future clinical studies.

[Amendment]

Thank you for comments to improve our manuscript. As suggested, we have added to the study limitation as follows:

“Lastly, this study needs more detailed mechanical exploration, the blocking study of the related protein in vivo, pharmacokinetics, investigation of different routes of administration and toxicity examinations in nanaomycin K. Further such studies will be undertaken in the next paper.” (line378-381)

In addition, we had a specialist proofread the English text, as it was pointed out to us that the English was difficult to understand.

Reviewer 2 Report (Previous Reviewer 2)

The manuscript has been greatly improved after revision.

Author Response

The manuscript has been greatly improved after revision.

[Amendment]

Thank you for comments to improve our manuscript. The comments have helped us significantly improve the paper.

Reviewer 3 Report (Previous Reviewer 1)

Authors answered clearly on each point I asked for, and they performed additional experiments by which they supported their results. Therefore, I recommend acceptance of this paper for the journal, after very small changes in the manuscript I listed here bellow:

2.4. Western blotting

I don’t see the defference between these antibodies. Please correct it. HRP- 150 conjugated secondary antibodies (Anti-IgG (H+L chain) (Mouse) pAb-HRP or Anti-IgG 151 (H+L chain) (Mouse) pAb-HRP (MBL, Nagoya, Japan))

2.5. Animal experiments

I suggest authors that instead of DMSO write: “The solvent used for administration to the animals was PBS, with a dosage of 80μL. The proportion of DMSO administered to the control group was 12%.”

2.6. Immunohistochemical staining

I suppose authors used on HRP-conjugated secondary antibody. Therefore should be written antibody, instead of antibodies.

2.7. Immunohistochemical analysis

After “The percentage of stained cells…” I would add “(frequency score)”.

4. Discussion

In the last paragraph where you wrote about limitations instead of sentence you wrote, you can write: “Fourth, the number of experimental animals used in in vivo experiments could be higher”. It sounds better.

Author Response

Authors answered clearly on each point I asked for, and they performed additional experiments by which they supported their results. Therefore, I recommend acceptance of this paper for the journal, after very small changes in the manuscript I listed here bellow:

2.4. Western blotting

I don’t see the defference between these antibodies. Please correct it. HRP- 150 conjugated secondary antibodies (Anti-IgG (H+L chain) (Mouse) pAb-HRP or Anti-IgG 151 (H+L chain) (Mouse) pAb-HRP (MBL, Nagoya, Japan))

[Amendment]

Thank you for comments to improve our manuscript. We have revised the relevant part as follow:

“HRP-conjugated secondary antibodies (Anti-IgG (H+L chain) (Mouse) pAb-HRP or Anti-IgG (H+L chain) (Rabbit) pAb-HRP (MBL, Nagoya, Japan))” (line149-150)

2.5. Animal experiments

I suggest authors that instead of DMSO write: “The solvent used for administration to the animals was PBS, with a dosage of 80μL. The proportion of DMSO administered to the control group was 12%.”

[Amendment]

Thank you for comments to improve our manuscript. We have revised the relevant part as follow:

“The solvent used for administration to the animals was PBS, with a dosage of 80μL. The proportion of DMSO administered to the control group was 12%.” (line161-162)

2.6. Immunohistochemical staining

I suppose authors used on HRP-conjugated secondary antibody. Therefore should be written antibody, instead of antibodies.

[Amendment]

Thank you for comments to improve our manuscript. I have revised it as you suggested. (line176)

2.7. Immunohistochemical analysis

After “The percentage of stained cells…” I would add “(frequency score)”.

[Amendment]

Thank you for comments to improve our manuscript. I have revised it as you suggested. (line185)

4. Discussion

In the last paragraph where you wrote about limitations instead of sentence you wrote, you can write: “Fourth, the number of experimental animals used in in vivo experiments could be higher”. It sounds better.

[Amendment]

Thank you for comments to improve our manuscript. We have revised the relevant part as follow:

“Then, the number of experimental animals used in in vivo experiments was not large.” (line373-374)

Round 2

Reviewer 1 Report (Previous Reviewer 3)

This revised manuscript investigates the anticancer activity of nanaomycin K, a novel compound extracted from Streptomyces sp., in castration-resistant prostate cancer (CRPC) and non-CRPC cell lines. Although the study provides interesting insights into the potential therapeutic effects of nanaomycin K, there are several concerns that need to be addressed for the manuscript to be considered for publication. Please find my major concerns below:

1. The study lacks proper controls in the experiments. The authors should include negative and positive control treatments, such as untreated cells or cells treated with known anticancer drugs, to validate the effects of nanaomycin K. Additionally, the authors should consider including non-malignant prostate cell lines as controls to evaluate the selectivity of nanaomycin K for cancer cells.

2. The mechanism of action of nanaomycin K remains unclear. The authors should perform additional experiments to strengthen their mechanistic insights into how nanaomycin K inhibits growth and migration in CRPC cells. This could include performing knockdown or overexpression experiments of the proposed target proteins (Ras, Slug, and MAPK), as well as investigating other potential targets or pathways involved in the compound's effects.

3. The in vivo experiments should be better described. Representative tumor mass: Presenting the average tumor mass of control and treated groups, along with a clear graphical representation, would provide a better understanding of the efficacy of nanaomycin K in inhibiting tumor growth in vivo. The authors should include a figure showing the tumor mass in each group at the end of the study, and the differences should be statistically analyzed. Furthermore, administration route and dosing: The authors should specify the route of administration of nanaomycin K (e.g., intravenous, intraperitoneal, or oral) and provide information about the dosing regimen, including the dose, frequency, and duration of treatment.

4. The authors should provide more evidence for the specificity of nanaomycin K's effects on CRPC cells. The author should investigate whether nanaomycin K has similar effects on non-cancerous prostate cells or other types of cancer cells. This would help to clarify the selectivity of the compound for CRPC.

5. Toxicity assessment: The authors claim that nanaomycin K safely inhibited tumor growth in vivo, but they do not provide any data on its toxicity or side effects. A thorough evaluation of the compound's toxicity is necessary to determine its potential for clinical application.

Minor concerns:

1. The manuscript would benefit from better organization and clarity. Several sections, such as the introduction and discussion, appear to be too brief and could be expanded upon to provide more context and interpretation of the results.

2. The manuscript would benefit from a more comprehensive discussion of the results. The authors should contextualize their findings within the existing literature on CRPC therapeutics and discuss the potential clinical implications of their work.

3. The authors should improve the quality and presentation of the figures. This includes providing high-resolution images, ensuring consistent formatting, and adding error bars to indicate the variability of the data.

4. The authors should address the limitations of their study, such as the potential off-target effects of nanaomycin K, the applicability of the findings to other cancer types, and the potential challenges in translating these findings to clinical settings.

Overall, the study shows potential but requires additional experiments and a more in-depth analysis to substantiate the claims and support the conclusions.

Author Response

This revised manuscript investigates the anticancer activity of nanaomycin K, a novel compound extracted from Streptomyces sp., in castration-resistant prostate cancer (CRPC) and non-CRPC cell lines. Although the study provides interesting insights into the potential therapeutic effects of nanaomycin K, there are several concerns that need to be addressed for the manuscript to be considered for publication. Please find my major concerns below:

1. The study lacks proper controls in the experiments. The authors should include negative and positive control treatments, such as untreated cells or cells treated with known anticancer drugs, to validate the effects of nanaomycin K. Additionally, the authors should consider including non-malignant prostate cell lines as controls to evaluate the selectivity of nanaomycin K for cancer cells.

[Amendment]

Thank you for comments to improve our manuscript. We have added our amendments to study limitation in the manuscript.  (page 12, line 388-389)

2. The mechanism of action of nanaomycin K remains unclear. The authors should perform additional experiments to strengthen their mechanistic insights into how nanaomycin K inhibits growth and migration in CRPC cells. This could include performing knockdown or overexpression experiments of the proposed target proteins (Ras, Slug, and MAPK), as well as investigating other potential targets or pathways involved in the compound's effects.

[Amendment]

Thank you for comments to improve our manuscript. We have added our amendments to study limitation in the manuscript.  (page 12, line 386-387)

3. The in vivo experiments should be better described. Representative tumor mass: Presenting the average tumor mass of control and treated groups, along with a clear graphical representation, would provide a better understanding of the efficacy of nanaomycin K in inhibiting tumor growth in vivo. The authors should include a figure showing the tumor mass in each group at the end of the study, and the differences should be statistically analyzed. Furthermore, administration route and dosing: The authors should specify the route of administration of nanaomycin K (e.g., intravenous, intraperitoneal, or oral) and provide information about the dosing regimen, including the dose, frequency, and duration of treatment.

[Amendment]

Thank you for comments to improve our manuscript. The route of administration is intratumorally and the dose is 80 µL. The frequency of administration is once per measurement day and the duration of administration is 5 days. These information were added in the manuscript. (page 5, line 165-167)

4. The authors should provide more evidence for the specificity of nanaomycin K's effects on CRPC cells. The author should investigate whether nanaomycin K has similar effects on non-cancerous prostate cells or other types of cancer cells. This would help to clarify the selectivity of the compound for CRPC.

[Amendment]

Thank you for comments to improve our manuscript. We have added our amendments to study limitation in the manuscript. (page 12, line 383-384)

5. Toxicity assessment: The authors claim that nanaomycin K safely inhibited tumor growth in vivo, but they do not provide any data on its toxicity or side effects. A thorough evaluation of the compound's toxicity is necessary to determine its potential for clinical application.

[Amendment]

Thank you for comments to improve our manuscript. We have added our amendments to study limitation in the manuscript. (page 12, line 392-395)

Minor concerns:

1. The manuscript would benefit from better organization and clarity. Several sections, such as the introduction and discussion, appear to be too brief and could be expanded upon to provide more context and interpretation of the results.

[Amendment]

Thank you for comments to improve our manuscript. Additions have been made to the introduction and discussion to improve the paper.  (page 3, line 75-78; page 12, line 373-382)

2. The manuscript would benefit from a more comprehensive discussion of the results. The authors should contextualize their findings within the existing literature on CRPC therapeutics and discuss the potential clinical implications of their work.

[Amendment]

Thank you for comments to improve our manuscript. The discussion has been expanded to include the relationship with currently used CRPC drugs. (page 12, line 373-382)

3. The authors should improve the quality and presentation of the figures. This includes providing high-resolution images, ensuring consistent formatting, and adding error bars to indicate the variability of the data.

[Amendment]

Thank you for comments to improve our manuscript. As suggested, we have made changes to the figure, such as standardizing the font.

4. The authors should address the limitations of their study, such as the potential off-target effects of nanaomycin K, the applicability of the findings to other cancer types, and the potential challenges in translating these findings to clinical settings.

[Amendment]

Thank you for comments to improve our manuscript. We have added our amendments to study limitation in the manuscript. (page 12, line 382-395)

Overall, the study shows potential but requires additional experiments and a more in-depth analysis to substantiate the claims and support the conclusions.

[Amendment]

Following your advice, we have added graphs, experiments, background and discussion from the original experiment to support our claims and conclusion. And experiments that are difficult to perform are mentioned in the limitations section.

In addition, the additional revisions are highlighted in light blue in the manuscript.

This manuscript is a resubmission of an earlier submission. The following is a list of the peer review reports and author responses from that submission.

Round 1

Reviewer 1 Report

Authors investigated the anticancer activity of nanaomycin K, a novel compound extracted from Streptomyces sp. using treatment-sensitive and castration-resistant prostate cancer (CRPC) cell lines. They investigated the effect of nanaomycin K on cancer cell lines growth in vitro, as well as the effect on cells invasion, even I would rather say migration or motility (please see below in the comments). Additionally, they investigated protein level of epithelial-mesenchymal transition (EMT) markers which causes invasion, then transcription factors that induce EMT, and MAPK signaling molecules level. Moreover, in vivo efficacy study of nanaomycin K was performed, followed by immunohistochemistry (IHC) ex vivo studies of tumor sections for epithelial, MAPK and apoptosis markers.

Although, the research is designed appropriately by applying suitable and relevant methods, and besides there are a lot of points which should be clarified, quantified, and better explained for the final acceptance of the manuscript, there is potential major problem of choosing not relevant model of the cell line TRAMP-C2, which was used and in in vivo study. Authors stated that it is “Murine androgen-independent cell line”. However there are listed here below studies where it was proven that this cell line is androgen-dependent cell line:

“The GFP-tagged cells (TRAMP-C2-GFP) display the same growth characteristics as parental cells and are androgen-dependent in vivo, and in vitro.” and “It is well known that stromal-epithelial interactions are very important for androgen dependent prostate cancer. Thus, co-implanting TRAMP-C2 cells with prostate stroma obtained from a donor mouse provides the tumor cells with an environment which closely resembles orthotropic implantation.” [Supplemental figure 1; https://www.ncbi.nlm.nih.gov/pmc/articles/PMC3139688/]

“…the apoptotic response of the TRAMP-C2 cells to androgen ablation in vivo supports androgen dependence for survival.” [PMID: 15797254]

On ATCC webpage for TRAMP-C2 is written that they are “Androgen receptor expressing cells”. Also, it is written that “These cell lines represent various stages of cellular transformation and progression to androgen-independent metastatic disease that can be manipulated in vitro.” [https://www.atcc.org/products/crl-2731]

I kindly please authors to explain this issue, and if changing statement of TRAMP-C2 cell line can be still adequate for the story of the research.

Please see more in the attached Review.

Best regards!

Reviewer 2 Report

Hirata and collaborators studied the anti-tumor Effect of nanaomycin k, a compound produced by Streptomyces sp., on prostate cancer cell lines in Vitro and In Vivo.  Overall, I believe the authors provide a novel compound extracted from Streptomyces sp. using treatment-sensitive and treatment-resistant prostate cancer cell lines. However, I have some some comments that can improve the quality of the manuscript:

1 Nananomycin K is a new compound. The author should provide the structure and structure identification map of the new compound.

1 The author mentioned that nanaomycin K is a new compound, so the author should provide the structure and structure identification map of the new compound.

2 The author proved that the nanaomycin K induced apoptosis of CRPC cells through the Caspase-3 pathway. However, the author did not involve the content of nanaomycin K induced apoptosis in the conclusion on line 32-33.

3 The secondary title of the results section should write the experimental results, not just the test methods, such as Cell proliferation assay, Wound healing assay, Western blotting, Animal experiments, Immunohistochemical analysis.

4 The author mentioned in the conclusion that the nanaomycin K may be a MAPK inhibitor, but the experimental results of inhibiting MAPK phosphorylation by nanaomycin K alone are not enough to reach this conclusion. The author should increase the target gene knockdown expression experiment to further confirm.

Reviewer 3 Report

General comments to the Authors

Despite the positive reviews of the original versions of the manuscript, there are glaring weaknesses that significantly diminish enthusiasm for its potential clinical utility of anti-tumor effect of Nanaomycin K in prostate cancer carcinomas via the MAPK signaling pathway. First, the study lacks requisite statistical power and replication to reliably validate the accuracy and reproducibility of its results and conclusions. Second, these studies uncover any detail mechanism that how Nanaomycin K disrupt the MAPK signaling pathway cascade axis connection, confers inhibition effect of malignant prostate cancer carcinomas metastasis and chemoresistance, and potentially regulates prostate cancer stemness. Third, the study is largely confirmatory of a previously published study by Sci Rep. 2021 Apr 28;11(1):9217.; J Biosci Bioeng. 2020 Mar;129(3):291-295. therefore lacks significant novelty.